# Oxygen–Ozone Therapy in the Rehabilitation Field:State of the Art on Mechanisms of Action, Safety andEffectiveness in Patients with Musculoskeletal Disorders

**DOI:** 10.3390/biom11030356

**Published:** 2021-02-26

**Authors:** Alessandro de Sire, Francesco Agostini, Lorenzo Lippi, Massimiliano Mangone, Simone Marchese, Carlo Cisari, Andrea Bernetti, Marco Invernizzi

**Affiliations:** 1Department of Medical and Surgical Sciences, University of Catanzaro “Magna Graecia”, 88100 Catanzaro, Italy; 2Department of Anatomy, Histology, Forensic Medicine and Orthopedics, Sapienza University, 00185 Rome, Italy; francescoagostini.ff@gmail.com (F.A.); massimiliano.mangone@uniroma1.it (M.M.); marchese.simone@gmail.com (S.M.); andrea.bernetti@uniroma1.it (A.B.); 3Department of Health Sciences, University of Eastern Piedmont, 28100 Novara, Italy; lorenzolippi.mt@gmail.com (L.L.); cisari50@gmail.com (C.C.); marco.invernizzi@med.uniupo.it (M.I.); 4Infrastruttura Ricerca Formazione Innovazione (IRFI), Azienda Ospedaliera SS. Antonio e Biagio e Cesare Arrigo, 15121 Alessandria, Italy

**Keywords:** ozone, oxygen ozone, oxygen–ozone therapy, pain management, rehabilitation, low back pain, neck pain, osteoarthritis, tendinopathy, fibromyalgia

## Abstract

In recent years, the interest in oxygen–ozone (O_2_O_3_) therapy application has considerably increased in the field of rehabilitation. Despite its widespread use in common clinical practice, the biochemical effects of O_2_O_3_ are still far from being understood, although its chemical properties seem to play a pivotal role in exerting its positive effects on different pathological conditions. Indeed, the effectiveness of O_2_O_3_ therapy might be partly due to the moderate oxidative stress produced by O_3_ interactions with biological components. O_2_O_3_ therapy is widely used as an adjuvant therapeutic option in several pathological conditions characterized by chronic inflammatory processes and immune over-activation, and most musculoskeletal disorders share these pathophysiological processes. The present comprehensive review depicts the state-of-the-art on the mechanisms of action, safety and effectiveness of O_2_O_3_ therapy in the complex scenario of the management of musculoskeletal disorders. Taken together, our findings suggest that O_2_O_3_ therapy seems to reduce pain and improve functioning in patients affected by low back pain and knee osteoarthritis, as reported by several studies in the literature. However, to date, further studies are warranted to clearly investigate the therapeutic effects of this promising therapy on other musculoskeletal disorders in the field of rehabilitation.

## 1. Introduction

Ozone (O_3_) is an inorganic molecule with allotrope properties consisting of three atoms of oxygen with a cyclic structure isolated for the first time in 1839 by Christian Friedrich Schönbein [1]. O_3_ is present in nature in the stratosphere but can also be produced artificially by subjecting diatomic oxygen (O_2_) to a high-voltage electrical discharge and appears colourless in gaseous form, with a characteristic smell [2,3,4,5,6].

In 1916, during the First World War, the O_3_ was used for its antimicrobial properties in wound healing at Queen Alexandra Military Hospital in London [7]. Since then, O_3_ has been utilized and extensively studied for over 100 years of medical history, with documented minimal side effects and some findings suggesting a therapeutic role in different medical fields [8]. O_3_ therapy was introduced as medical therapy in the 19th Century, with the first O_3_ generator patented by Nikola Tesla [8]. The introduction of specific and certified O_3_ generators allowed physicians to create an O_2_O_3_ mixture with precise concentrations in order to avoid the toxicity induced by excessive oxidative stress related to O_3_ high reactivity [8,9].

Nowadays, a medical mixture composed of O_2_ and O_3_ is produced by a medical generator from pure O_2_ passing through a high-voltage gradient (5–13 mV) but, unfortunately, it cannot be stored because of O_3_’s high instability (half-life of 40 min at 20 °C) [10]. Moreover, the medical gas mixture concentration should be composed of no less than 95% O2 and no more than 5% O_3_ [9]. In recent years, there has been a growing interest in the scientific literature about the biochemical properties of the O_2_O_3_ mixture in order to better understand the basic mechanisms of action underpinning its systemic effects on human blood and tissues [10,11,12,13]. Several papers suggested relevant medical features of O_3_, including bactericidal and virucidal properties, inflammatory modulation and circulatory stimulation, with multiple applications in several medical fields including wound healing, ischemic disorders, infections and chronic inflammatory conditions like musculoskeletal disorders [14,15,16,17,18,19,20,21].

Musculoskeletal disorders include several pathological conditions characterized by a consistent prevalence and burden in terms of sanitary costs and residual disability, mainly related to musculoskeletal pain [22,23]. In this context, Physical and Rehabilitation Medicine (PRM) is a specialist area of medicine focused on the complex management of functioning impairments and disabling sequelae related to several chronic conditions, and musculoskeletal disorders account for a significant portion of its clinical tasks [24]. In this scenario, conservative and mini-invasive treatment options are extremely functional to the PRM framework [24,25,26,27,28]; thus, the interest in O_2_O_3_ therapy application in this field has considerably increased in recent years [13].

However, to date, although many studies have suggested O_2_O_3_ therapy as an effective therapeutic option in the management of several musculoskeletal disorders [29,30,31], common agreement about the specific indications and treatment modalities is lacking. Therefore, the present review aims at describing the state-of-the-art regarding the mechanisms of action, treatment modalities and potential side effects of O_2_O_3_ therapy in the complex management of the most common musculoskeletal disorders in the PRM field.

## 2. Oxygen–Ozone Therapy: Mechanisms of Action

In the last four decades, several hypotheses have been proposed to clarify the mechanisms underpinning the antioxidant, antalgic, anti-inflammatory and immunomodulatory actions of medical O_2_O_3_ mixture. However, despite its various and heterogenous medical applications, the biochemical effects of the O_2_O_3_ mixture are far from being understood in detail, even though its chemical properties seem to play a pivotal role in exerting its positive effects in different pathological conditions [13,29,30,31,32,33,34,35].

O_3_ is considered one of the most powerful oxidizing molecules in nature, although, at high concentrations, it rapidly decays into ordinary oxygen [36,37]. As reported by Bocci and colleagues in different studies [1,2,3,4,5,6,9,10,17,37,38,39] in human fluids and tissues O_3_ rapidly reacts with water and polyunsaturated fatty acids (PUFA), creating, respectively, hydrogen peroxide (H_2_O_2_) and a mixture of lipid ozonation products (LOP), mainly composed by 4-HNE (from omega-6 PUFA) and 4-HHE (trans-4 hydroxy-2-hexenal from omega-3 PUFA).

H_2_O_2_ is considered the fundamental reactive oxygen species (ROS), and acts as an ozone messenger [40]. However, other ROS have been identified as a product of ozone reactions, such as superoxide ion and hydroxyl radical (OH^−^) [39]. Given the role of the signal transduction, the previous concept that ROS are always harmful has recently been revised and replaced by the latest evidence. Thus, ROS could be considered mediators of the host defense and immune responses [39,40]. However, given their high reactivity, ROS could damage crucial cell components, and their generation should be precisely calibrated, considering their extremely short lifetime (a few seconds).

In human tissues, the moderate oxidative stress caused by ROS is nullified by endogenous radical scavengers, such as superoxide dismutase, glutathione peroxidase, catalase, and NADPH quinone-oxidoreductase [12,41,42]. It has been shown that small and repeated oxidative stresses could induce the activation of transcriptional factor mediating nuclear factor-erythroid 2-related factor 2 (Nrf2), a domain involved in the transcription of antioxidant response elements (ARE) [12].

Previous studies on human cytosol showed that Nrf2 is usually bound with Kelch-like ECH-associated protein 1 (Keap-1), creating an inactive complex in the intracellular space [43,44]. It has been proposed that mild oxidative stress might promote Nrf2 release from this complex and its migration into the nucleus, where binding with the Maf protein could promote the transcription of different ARE on DNA [42]. This pathway is described in detail by Figure 1.

Thus, through repeated mild oxidative stresses, O_3_ could induce the upregulation of Nrf2, conditioning human cells to transcript different AREs. This could result in a better response to pathological radical stress, common in most chronic inflammatory diseases [12,42]. Intriguingly, Nrf2 also seems to play an important role in the intracellular signalling pathways of inflammation. In this regard Li et al. [44] showed that the activation of Nrf2-antioxidant signalling might attenuate NF-κB, a key regulator of inflammatory response and muscle atrophy, as underlined by several in vivo studies [45,46,47,48,49,50].

Furthermore, studies on hepatic and heart ischemia-reperfusion suggested that the inflammatory response might be directly downregulated by the suppression of crucial inflammatory mediators and cytokines like IL-6, IL-8 and TNF-a [51,52,53]. Similarly, low doses of O_3_ might also have a role in the regulation of prostaglandin synthesis, the release of bradykinin, and in increasing secretions of macrophages and leukocytes [54,55].

Several studies have assessed the effects of O_2_O_3_ therapy in different pathological conditions related to musculoskeletal pain [29,30,31,32,33,34,35]. It is widely accepted that pain is a common symptom related to the inflammation process, and O_2_O_3_ therapy might play a key role not only in the management of inflammation but also in nociceptive perception and modulation [29,30,31,32,33,34,35]. Hence, an increase in serotonin and endogenous opioids has been shown after O_2_O_3_ administration, and these antioxidant molecules could induce pain relief by stimulating antinociceptive pathways [12,56,57,58,59,60].

Hypoxia and vascularization impairment are a common pathological feature in muscle wasting and musculoskeletal disorders; thus, as suggested by Clavo et al. [61], the potential role of O_2_O_3_ therapy in hypoxic tissues may be related to an increased production of nitric oxide, adenosine and prostaglandins, with a resulting positive role in the vasodilatation process. Lastly, another study by Bocci et al. [4] reported a possible direct effect of O_2_O_3_ in human blood, related to the shift to the right in the oxyhaemoglobin dissociation curve due to the increase in 2,3-diphosphoglycerate production and lipid peroxidation.

Taken together, all these findings suggested that O_2_O_3_ might exert its positive effects in several tissues due to the moderate oxidative stress produced by O_3_ interactions with biological components. However, it should be noted that “the thin red line” between O_3_′s beneficial effects and toxicity could be related to the strength of the oxidative stress.

However, up to now, further studies are warranted to better characterize the biochemical and biomolecular modifications induced by O_2_O_3_ and to deeply understand the complex regulation of different human pathways responsible for the therapeutic effects of this innovative and promising intervention.

## 3. Oxygen–Ozone Therapy: Indications and Contraindications

O_2_O_3_ therapy is widely used as an adjuvant therapeutic option in several pathological conditions characterized by chronic inflammatory processes and immune overactivation and most musculoskeletal disorders share these two pathophysiological processes [12]. In this context, several studies in the literature suggested an effective role of O_2_O_3_ in the management of common vertebral column degenerative diseases. Several studies performed in low back pain (LBP) patients showed good perspectives for the conservative treatments of disc herniation or protrusion and in case of failed back surgery syndrome [62,63,64,65]. In this scenario, O_2_O_3_ could be used with an indirect and minimally invasive approach, targeting the paravertebral muscles corresponding to the metamer of the herniated disc [62,63].

Another site worthy of interest for O_2_O_3_ therapy is the knee joint. A recent systematic review performed by Sconza et al. [66] reported that knee pain could be effectively reduced after O_2_O_3_ intra-articular administration in patients affected by knee osteoarthritis (KOA). Similarly, disorders at the tendon level are another potential target for O_2_O_3_ therapy that have recently gained interest in the scientific literature. A recent randomized controlled trial (RCT) assessed the efficacy of O_2_O_3_ therapy in patients with shoulder impingement, showing that it might be considered an intriguing alternative treatment in case of contraindication to corticosteroids [36]. Moreover, Ulusoy et al. [35] reported positive results after O_2_O_3_ injective treatment in patients affected by lateral chronic epicondylitis not respondent to conventional treatment. Furthermore, promising results have been reported even in rheumatic diseases, where O_2_O_3_ rectal insufflations or autohemotherapy showed a high safety profile and encouraging positive results in fibromyalgia [59]. The most common sites of action and administration modalities for O_2_O_3_ therapy are depicted in Figure 2.

Like any other therapeutic intervention, O_2_O_3_ is not devoid of potential side effects. O_2_O_3_ therapy contraindications are mainly related to the antioxidant characteristics of this mixture [5,9,10]. In more detail, glucose-6-phosphate dehydrogenase deficiency is the main contraindication, given the red blood cell breakdown triggered by the oxidative stress induced by the O_3_. Other contraindications could include pregnancy (albeit that this is a relative contraindication), uncontrolled hyperthyroidism, severe cardiovascular diseases and heart failure.

O_2_O_3_ concentrations should be set to a specific range to ensure safety; however, patients might present a sensation of heaviness at the injection site that spontaneously decreases in a few minutes. On the contrary, other adverse effects might be related to an incorrect administration technique, including vagal crisis, pain, hematoma in the injection site, local infections, and even death [67,68]. In this context, ultrasonography has been recently proposed as a non-invasive and real-time technique that could guide O_2_O_3_ injection, allowing the physician to precisely target the area of interest and to monitor the gas spreading in the tissues, in order to reduce the occurrence of adverse events [69]. To date, despite the widespread literature, evidence about O_2_O_3_ therapy efficacy in musculoskeletal pain management is lacking. This might be due to recurrent bias, low quality studies and the heterogeneity in treatment protocols and administration methods. Therefore, in the following sections of this review, we will summarize the current evidence on the therapeutic effects of O_2_O_3_ in the most common musculoskeletal disorders in the PRM field.

## 4. Oxygen–Ozone Therapy in Low Back Pain Treatment

O_2_O_3_ therapy has been proposed for several years as an effective treatment in patients affected by LBP due to intervertebral disc herniation [70,71,72]. A recent systematic review and metaregression [63] highlighted the positive effects in terms of pain reduction and functioning improvement in these patients. The O_2_O_3_ might exert its action in reducing LBP with a coupled mechanical and anti-inflammatory effect. The oxidizing action might break glycosaminoglycan chains in the nucleus pulposus, reducing their ability to retain water, thus decreasing the size of the herniated position and helping to reduce hernial conflict [12,42]. O_2_O_3_ might have also an effect on the inflammatory cascade by altering the breakdown of arachidonic acid into inflammatory prostaglandins [54,55]. Lastly, it stimulates the fibroblastic activity, boosting the deposition of collagen and the initiation of the repairing process at tissue level [73].

At present, the most widely used O_2_O_3_ technique used to treat patients affected by LBP is the intramuscular–paravertebral injection. However, even intradiscal and intraforaminal O_2_O_3_ therapy have been recently proposed and investigated for the complex management of LBP in the rehabilitation setting.

### 4.1. Intramuscular–Paravertebral Oxygen–Ozone Therapy in Low Back Pain Treatment

This procedure commonly consists of a paravertebral intramuscular injection of 20 mL of O_2_O_3_ into the paravertebral muscles at the level of the metameres of the herniated disc (half dose per side). The threshold level varies between 15 and 30 μg/mL, depending on the individual’s antioxidant capacity, which is directly related to older age, obesity, and alteration of the physiological sagittal spine curvatures [4,63].

Furthermore, an adequate differential diagnosis is mandatory to exclude other musculoskeletal conditions with a similar symptomatology (e.g., lumbar facet syndrome, piriformis syndrome, trochanteric bursitis, sacroiliac joint pain, etc.). It should also be taken into consideration that pain might spontaneously regress in up to 90% of patients with acute-onset herniated disc, Indeed, a natural, positive progression of the herniated disc and the spontaneous reduction in herniated volume over time might be confounding factors. Starting from these considerations, a follow-up evaluation with an adequate assessment of outcome measures (e.g., pain, functioning, health-related quality of life (HRQoL)) should be performed at three months after the end of O_2_O_3_ therapy [63].

A multicenter randomized, double-blind, simulated therapy-controlled trial performed by Paoloni et al. in 2009 [29] on 60 patients with acute LBP due to lumbar disc herniation assessed the effectiveness of intramuscular–paravertebral injections of O_2_O_3_. A significant difference was observed between the two groups in the percentage of cases who had become pain-free. The authors concluded that in case of non-responsivity to conservative approaches, intramuscular lumbar paravertebral O_2_O_3_ injections seem to be a safe and effective way to relieve pain and reduce disability and the intake of analgesic drugs in LBP patients.

Similarly, in 2014, Apuzzo et al. [70] assessed the efficacy of O_2_O_3_ therapy in back pain rehabilitation, comparing three groups of patients suffering from chronic back pain associated with disc herniations. Patients underwent three different treatments: intramuscular–paravertebral O_2_O_3_ injections, global postural re-education, or a combination of O_2_O_3_ injective therapy and rehabilitation. At the end of the treatment protocol, pain severity (assessed by Visual Analogue Scale, VAS) was lower in patients who performed O_2_O_3_ therapy than in those who undergone global postural re-education only.

In 2018, Niu et al. [71] investigated the therapeutic effect of low, medium, and high concentrations of O_2_O_3_ on trauma-induced lumbar disc herniation. They enrolled 80 patients and divided them into a control group and a low O_2_O_3_ (20 μg/mL), medium O_2_O_3_ (40 μg/mL) and high O_2_O_3_ (60 μg/mL) group. The authors used a CT scan and an enzyme-linked immunosorbent (ELISA) assay to detect serum IL-6 levels, SOD activity, IgM and IgG levels upon admission and at 6 and 12 months after follow-up. All patients showed a CT disc volume reduction at 6- and 12-month follow-up. Moreover, the study demonstrated that O_2_O_3_ concentration of 40 μg/mL provided the optimal anti-inflammatory treatment efficacy, while low concentrations of O_2_O_3_ (20 μg/mL and 40 μg/mL) reduced the serum IL-6, IgG, and IgM concentrations, showing relevant anti-inflammatory effects. On the contrary, high concentrations of O_2_O_3_ (60 μg/mL) increased the serum IL-6, IgG, IgM expression, exacerbating pain and proinflammatory effects in these patients.

A recent paper by Özcan et al. [72] assessed the effects of paravertebral O_2_O_3_ therapy (every 7 days for a total of 6 injections) in 122 patients with LBP. The authors reported a statistically significant improvement in Visual Analogue Scale and Oswestry Disability Index (ODI) scores at first month follow-up, concluding that paravertebral O_2_O_3_ is a reliable and effective treatment in the complex management of LBP related to lumbar disc herniations. Similarly, Biazzo et al. [73] investigated the role of paravertebral injective O_2_O_3_ therapy in 109 patients affected by LBP, showing that 79% of patients had a significant pain and back disability reduction, corroborating the hypothesis that intramuscular O_2_O_3_ injections can be considered a safe, reliable and effective therapeutic intervention in LBP management.

Accordingly, our study group [30] has recently described the effects of O_2_O_3_ therapy in a 68-year-old woman with an acute episode of severe LBP, not respondent to opioids. The patient presented with several disc herniations (L3–L4, L4–L5, and L5–S1 levels) and a concomitant sacral insufficiency fracture and was treated with intramuscular–paravertebral injections of O_2_O_3_ for 4 weeks (once per week), using a O_3_ concentration of 20 μg/mL (5 mL in L4–L5 zone and 5 mL in L5–S1 zone, bilaterally). We found a consistent reduction in pain (assessed by Numeric Pain Rating Scale, NPRS) and an improvement in HRQoL (assessed by Short Form 12-Item Health Survey, SF12) at 1 week after the first injection (NPRS: 5.5 vs. 8.5; SF-12 Physical Composite Score, PCS: 30.4 vs. 25.2), confirmed at the one-month follow-up visit (NPRS: 1 vs. 8.5; SF-12 PCS: 45.0 vs. 25.2).

Furthermore, another recent case report performed by Bellomo et al. [74] evaluated the effects of percutaneous injections of O_2_O_3_ in a 73-year-old woman affected by acute LBP due to a facet joint syndrome. The patient was treated with O_2_O_3_ (1 injection at 20 μg/mL per week for 3 weeks) under ultrasound guidance, and after a break of 1 week, she also performed exercises for water rehabilitation (twice a week for 4 weeks) with intriguing results.

To date, three systematic reviews [75,76,77] have investigated the effects of percutaneous injections of O_2_O_3_ in patients affected by LBP. More in detail, in 2011, Magalhaes et al. [75] systematically reviewed the literature from 1966 to September 2011, focusing on the efficacy of percutaneous injections of O_2_O_3_ in LBP due to disc herniations. Despite the lack of precise LBP diagnosis and the frequent use in some studies of mixed therapeutic agents, the authors concluded that O_2_O_3_ therapy seemed to yield positive results and low morbidity rates when applied percutaneously for the treatment of chronic LBP.

Costa et al. [76] in 2018 conducted a systematic review aimed at investigating both effectiveness and safety of O_2_O_3_ therapy in LBP patients with lumbar disc herniations. As previously described, most of the reviewed studies were of poor quality; however, they all reported a consistent pain reduction after O_2_O_3_ therapy and few adverse events (mostly minor).

Lastly, in 2020, Barbosa et al. [77] performed a cross-sectional review using the PubMed, LILACS and Scopus databases, which aimed at addressing the efficacy and adverse events occurrence of O_2_O_3_ in the treatment of LBP. The authors concluded that the use of intramuscular–paravertebral O_2_O_3_ in LBP patients could be suggested as an effective and safe intervention, especially when compared to surgery.

In conclusion, intramuscular–paravertebral O_2_O_3_ therapy seems to be safe, reliable and effective to reduce pain in patients affected by LBP not responding to anti-inflammatory/analgesic drugs. Intramuscular–paravertebral O_2_O_3_ might be considered a promising technique that could be integrated as part of the multidisciplinary rehabilitation management of these patients. However, to date, medical guidelines about O_2_O_3_ use in LBP are still warranted to improve the knowledge on the specific indications and therapeutic modalities of this promising intervention.

### 4.2. Intradiscal and Intraforaminal Oxygen–Ozone Therapy in Low Back Pain Treatment

In the past, several studies have investigated the role of intradiscal and intraforaminal ozone applications in pathological spinal disc conditions. However, the need for a fluoroscopy or tomographic guide has limited the spread of this technique in the common clinical rehabilitative setting [78]. This therapeutic approach consists of an intradiscal injection of high concentrations O_2_O_3_, aimed at reducing intradiscal pressure through glycosaminoglycan lysis, proteoglycan reduction and disc dehydration [79].

In 2004, in an observational study on 2200 participants, Muto et al. [80] assessed the safety and efficacy of intradiscal and intraforaminal O_2_O_3_ injection in patients with LBP or sciatica due to a herniated disk. The authors did not record any neurological or infective complications, while they observed a significant success rate in 80% of patients at short-term follow-up (six months). In more detail, the data obtained suggested a lower response rate in patients affected by a calcified herniated disk, spinal canal stenosis, a recurrent herniated disk or a small descending herniated disk of the lateral recess.

On the other hand, the conservative management of LBP has been deeply studied by Bonetti and colleagues [81]. They performed a large RCT involving 306 patients assessing the efficacy of intraforaminal ozone infiltration compared to peri-radicular steroid infiltrations. Intriguingly, significant differences in terms of pain relief were found between groups in patients with disc diseases, suggesting the positive effects of ozone in these specific patients.

Similarly, a RCT performed by Gallucci et al. [82] compared the effects of intradiscal and intraforaminal injections of steroid and O_2_O_3_ versus the steroid injections only. After six months of follow-up, the authors reported a significantly higher success rate in the combined treatment group compared to the steroid alone group (74% vs. 47%: *p* < 0.05).

In 2008, Muto et al. [83] conducted a large sample observational study involving 2900 patients with LBP and sciatica treated with O_2_O_3_ chemonucleolysis procedures. The authors concluded that intradiscal intraforaminal O_2_O_3_ therapy can be considered a safe and cost-effective technique with positive results in LBP and sciatica patients.

In 2013, Zhang et al. [84] showed a VAS score reduction from 7.68 to 2.17 (*p* < 0.05) after intradiscal and intraforaminal injection of ozone concentrations of 25–30 µg/mL in patients with low back pain and radicular pain. Thus, the authors suggested that O_2_O_3_ mixture could be considered as a first-choice treatment before surgery in patients with herniated disc nonresponsive to other conservative therapies.

Similarly, a prospective randomized double-blind trial by Perri et al. [85] performed on 517 patients compared intradiscal O_2_O_3_ injection with steroids injection. Interestingly, six months after the treatment, the authors reported a successful outcome in 80% of patients treated with both intradiscal ozone therapy and intraforaminal corticosteroid injection, while 31.5% of the control group treated with intraforaminal corticosteroid injection only.

In 2018, Elawamy et al. [86] assessed the effects of intradiscal injections of O_2_O_3_ in 60 patients with radicular leg pain and LBP, comparing different doses of the mixture. In particular, the treatment proposed was composed of 10 mL with concentration of 40 µg/mL O_3_ compared to 30 µg/mL O_3._ Although the authors did not underline significant differences between groups, the data showed a significant improvement in both the pain and functional outcomes after a single intradiscal injection of O_2_O_3_ in all the study groups. Thus, the authors concluded that intradiscal injection might be considered a very valuable maneuver to improve pain and disc herniation.

In 2020, a double-blinded RCT performed by Ercalik and Kilic [87] assessed the effects of a single intradiscal injection of 5 mL of O_2_O_3_ mixture containing 40 μg/mL O_3_, with or without intraforaminal injection of 8 mg of dexamethasone and 1 mL of 0.05% bupivacaine. The authors found a significant improvement in terms of pain, disability, and HRQoL in both groups after the mini-invasive procedure, without significant differences between the groups.

In conclusion, the available literature reported a high safety profile of intradiscal and intraforaminal oxygen–ozone administration. Although the need for a fluoroscopy or tomography guide could limit the feasibility of this treatment in the common rehabilitation setting, positive effects were reported when compared to other therapies such as steroid intraforaminal injection. Lastly, the beneficial effects of the treatment could be related to the size and localization of the herniated disk, patients’ characteristics, and O_2_O_3_ volume and concentration. Therefore, further research should focus on assessing the best approach to improve the treatment effectiveness.

## 5. Oxygen–Ozone Therapy in Knee Osteoarthritis Treatment

KOA is frequent condition, characterized by joint pain and periarticular muscle weakness, with subsequent loss of function, increased disability, lower performance in the activities of daily living and a relevant reduction of HRQoL [88,89]. Thus, an early diagnosis of OA is needed to better define the adequate therapeutic approach [90], consisting of nonsurgical treatments such as physical exercise [91] and pharmacological treatments (e.g., acetaminophen, NSAIDs, and opioids) [92].

Among nonpharmacological approaches, intra-articular injectable forms of hyaluronic acid (HA) are frequently used in the PRM clinical practice to improve the viscoelastic properties of the synovial fluid in patients affected by KOA and are considered a safe and well-tolerated intervention [88,93,94]. However, according to several practice guidelines [92,95], intra-articular HA administration is still considered as a second-line approach in patients affected by KOA not respondent to acetaminophen and/or NSAIDs. Thus, there has been a growing interest in the literature regarding the use of other nonpharmacological rehabilitative approaches in the management of KOA, including focal vibration [96], radiofrequency ablation of genicular nerves [97], and intra-articular O_2_O_3_ therapy [31,98,99,100,101,102,103,104].

In more detail, in 2016, Giombini et al. [98] firstly compared the intra-articular injection of O_2_O_3_, HA, and a combined therapy of both in patients with KOA. They reported significant (*p* < 0.05) pain reduction (VAS) and disability (Knee Injury and Osteoarthritis Outcome Score Analysis, KOOS) after all three approaches (O_2_O_3_, HA, and their combination) at the end of treatment cycle and at the two-month follow-up evaluation. Moreover, the combination of O_2_O_3_ and HA treatment led to a significantly better outcome, especially after two months.

In 2018, another RCT [103], compared the efficacy of O_2_O_3_ therapy versus HA intra-articular injection in patients affected of KOA. The authors randomly allocated the study participants into two groups: an O_2_O_3_ group and a HA group, both treated with a protocol consisting of three weekly injections. The total Western Ontario and McMaster Universities Arthritis Index (WOMAC) score decreased from 40.8 ± 9.8 to 20.4 ± 4.9 (*p* < 0.01) in the O_2_O_3_ group and from 38.5 ± 7.9 to 17.1 ± 4.2 (*p* < 0.01) in the HA group. The authors concluded that knee pain, stiffness, and function significantly improved in both groups, without any difference between the two interventions at six-month follow-up.

Lastly, our group [31] recently performed a randomized single-blind study, including 42 patients randomly allocated to receive a cycle of O_2_O_3_ therapy (22 patients) and a cycle of hyaluronic acid (20 patients). Patients in both groups showed a significant reduction of VAS (*p* < 0.013) compared to baseline during both cycles. At follow-up evaluations, VAS was significantly lower in the HA group (*p* < 0.013). Thus, we concluded that intra-articular O_2_O_3_ might be comparable to HA in reducing pain in KOA patients, albeit that VAS was significantly lower in the HA group at both follow-up visits.

In addition to HA, O_2_O_3_ therapy has been compared with other therapeutical approaches, including corticosteroids, oral celecoxib and glucosamine [99,104]. A RCT performed by Babaei-Ghazani et al. [104] compared the effects of ultrasound-guided corticosteroid injections to O_2_O_3_ therapy for the treatment of KOA. The study included 62 patients suffering from knee pain in the last six months, who were divided into two groups: the first group was treated with 40 mg of triamcinolone and the second group with 10 cc of O_2_O_3_ injected in the knee joint with the support of ultrasound guidance. The authors concluded that both steroid and O_2_O_3_ injections might be effective in the management of KOA patients, and, although steroid injection showed an earlier improvement in knee pain, the effects of O_2_O_3_ seem to last longer.

In the same way, Feng et al. [99] compared the effectiveness of O_2_O_3_ therapy to anti-inflammatory drugs and nutraceuticals for the treatment of KOA. The study enrolled 76 patients and randomly assigned them into two groups: a control group undergoing oral celecoxib and glucosamine and another performing intra-articular O_2_O_3_ therapy. Their findings showed a greater pain reduction in the O_2_O_3_ group compared to the control group three weeks after the end of the treatment.

To date, there has only been one randomized, double blinded, placebo-controlled study [100] that investigated the efficacy of intra-articular O_2_O_3_ therapy in KOA patients. Lopes de Jesus et al. randomly divided 98 KOA patients into two groups: 63 patients in the O_2_O_3_ group (eight injections once a week) and 35 patients in the placebo group. After eight weeks of treatment, O_2_O_3_ was more effective than the placebo in terms of pain reduction (mean difference, MD, of VAS = 2.16; *p* < 0.003), ambulation (MD of Lequesne Index = 4.05, *p* < 0.001), and HRQoL (MD of general state of health 36-Item Short Form Health Survey = -3.38, *p* < 0.001). Moreover, only three patients (one in the O_2_O_3_ group and two in the placebo group) experienced minor adverse events (puncture accidents), suggesting a favourable safety profile of intra-articular O_2_O_3_.

To date, the effectiveness of intra-articular O_2_O_3_ therapy for KOA patients has been investigated in six systematic reviews and meta-analyses [67,101,102,103,104,105,106]. In more detail, two of them [101,102] aimed to compare the O_2_O_3_ therapy with HA injections in KOA patients.

Li et al. [101] performed a meta-analysis, including 4 RCTs with a total of 289 patients, concluding that intra-articular injection of HA was associated with a significant VAS reduction compared to O_2_O_3_ therapy at one month after treatment. Similarly, Javadi Hedayatabad et al. [102] confirmed these results, reporting a better performance of HA injections in terms of WOMAC improvement, albeit that there was no significant difference between HA and O_2_O_3_.

On the contrary, four recent systematic reviews and meta-analyses [66,103,105,106] reach different conclusions about O_2_O_3_ efficacy in KOA patients. Raeissadat et al. in 2018 [103] included 428 patients from 5 RCTs, who were divided in two groups: 53% in the O_2_O_3_ group and 47% in the control group. The authors concluded that the efficacy of intra-articular O_2_O_3_ therapy in pain reduction was significantly superior to the placebo and not statistically different from the other treatments (HA, dextrose, and air injection), suggesting a putative role of O_2_O_3_ in the nonsurgical short term (3–6 months) management of mild–moderate KOA.

Similarly, in 2019, Arias-Vàzquez et al. [105] confirmed in their meta-analysis that O_2_O_3_ therapy had a significant short-term effectiveness in reducing knee pain (d = −2.26, I2 = 97; *p* < 0.001) lasting on average between three and six months, and these results were also confirmed by another recent meta-analysis performed by Noori-Zadeh et al. [106].

Lastly, in 2020, Sconza et al. [66] performed a systematic review of RCTs about O_2_O_3_ treatment efficacy in KOA patients. A total of 11 studies involving 858 patients were included; however, none of the studies included reached a “good quality” standard. Thus, the authors concluded that no clear indications emerged from the comparison of O_2_O_3_ with other established treatments for KOA, but this technique was proven to be safe and effective in reducing pain in the short–middle term in these patients.

Taken together, all these findings suggested that intra-articular O_2_O_3_ therapy might be considered as a safe and effective treatment in the short–middle term (up to six months) in patients affected by KOA; however, further high-quality studies are warranted to improve the scientific knowledge regarding this promising conservative intervention.

## 6. Oxygen–Ozone Therapy in Other Musculoskeletal Disorders of Rehabilitative Interest

Several other musculoskeletal diseases might take advantage of the O_2_O_3_ therapy that is commonly used in the PRM clinical practice. However, only a few papers have investigated the effects of O_2_O_3_ therapy on other musculoskeletal disorders leading to disability (i.e., cervical pain, tendinopathies, and fibromyalgia).

Cervical pain is commonly due to cervical spondylolysis or osteoarthritis, a chronic degenerative condition inducing changes in the bones, intervertebral discs and/or joints connected to the neck. Cervical osteoarthritis might induce progressive functioning impairment and disability and conservative approaches, including rehabilitative exercise, are the most recommended therapeutic interventions [107].

Among the pool of conservative interventions, O_2_O_3_ therapy is widely used in the PRM common clinical practice in patients with cervical pain, albeit that there is still little evidence supporting its efficacy. In 2018, Raeissadat et al. [34] firstly assessed the role of paravertebral O_2_O_3_ injection in 72 patients affected by chronic nonspecific cervical pain lasting more than three months despite conservative treatments and with active trigger points in the upper trapezius muscles. In this single-blinded study, they compared O_2_O_3_ therapy with other two interventions commonly used in clinical practice: 2 cc of lidocaine 2% injection in the trapezius trigger points and dry needling, respectively. All three of these interventions were significantly effective in terms of reducing VAS reduction and increasing pain pressure threshold after a four-week follow-up. However, O_2_O_3_ and lidocaine treatment showed superior, although not statistically different, results compared to dry needling group.

Furthermore, a retrospective study conducted by Lin and colleagues [108] assessed the effectiveness of ultrasound guided percutaneous O_2_O_3_ injections around the cervical dorsal root ganglions in patients affected by cervical pain due to zoster. The authors treated 30 subjects with zoster-associated pain with ultrasound-guided percutaneous O_2_O_3_ injection around the cervical dorsal root ganglion at the injured nerve level (C2-C8). Patient improvements in pain and neurologic function were evaluated during a follow-up period from one to three months, showing that percutaneous O_2_O_3_ injection around the dorsal root ganglion might be an effective intervention for treatment-resistant cases of zoster-associated pain at the cervical level.

Lastly, a recent study performed by Martinelli et al. [19] in 2020 evaluated the safety and effectiveness of intramuscular–paravertebral injections of O_2_O_3_ in 168 patients affected by cervicobrachial pain. The study participants received 12 cervical intramuscular injections of an O_2_O_3_ mixture (5 mL) with an O_3_ concentration of 16 μg/mL once a week, showing a significant pain reduction (*p* < 0.001) at one, two, three, four and five years of follow-up.

In conclusion, despite the few papers in the literature addressing this issue, intramuscular–paravertebral O_2_O_3_ injections might be considered as a promising conservative approach in patients affected by cervical pain, although the data about safety are lacking. It is mandatory that further prospective studies will better investigate the role of O_2_O_3_ in reducing pain and pain-related disability in patients suffering from cervical pain, with a special focus on the safety profile of this approach.

Acute and chronic pathological modifications at tendon level are a pool of musculoskeletal disorders that might involve physicians from different fields, including orthopedics, sports medicine specialists, and PRM specialists [109]. The conservative management of tendinopathies is the first line of intervention and is preferable, when possible, to surgery. In this scenario, O_2_O_3_ therapy is a promising conservative approach used in common clinical practice as an alternative to local corticosteroid treatment. However, to date, clear indications about the use of O_2_O_3_ therapy in patients affected by tendinopathies are lacking.

Ulusoy et al. [35], in 2019, performed a study in 80 patients with chronic lateral epicondylitis comparing the effectiveness of corticosteroid and O_2_O_3_ therapy on pain reduction. Corticosteroid injection was administered once a week for three times, while O_2_O_3_ injection was administered eight times (every three days). Both treatments significantly reduced pain without differences between groups. Moreover, O_2_O_3_ showed better results in terms of pain reduction than corticosteroids at three, at six and at nine months after the injections, suggesting that O_2_O_3_ could be considered an effective therapeutic option in the treatment of chronic lateral epicondylitis.

Furthermore, another musculoskeletal condition of rehabilitative interest is fibromyalgia, due to the high burden in terms of disability that affects a high percentage of these patients [110]. Fibromyalgia is a systemic pain syndrome characterized by widespread musculoskeletal pain accompanied by fatigue, sleep disturbance, and other symptoms as a result of the dysregulation of neurophysiological function [111]. Few findings are present in the literature about the use of O_2_O_3_ therapy in these patients [59,112,113].

Hidalgo-Tallòn J. et al. [112] in 2012, tested the effectiveness and tolerability of O_2_O_3_ therapy administered by rectal insufflation as a complementary intervention in the complex multidisciplinary fibromyalgia management. Thirty-six patients affected by fibromyalgia received 24 sessions of O_2_O_3_ therapy (200 mL of gas, at a concentration of 40 μg/mL) during a 12-week period with a significant reduction in fibromyalgia physical symptoms assessed with the Fibromyalgia Impact Questionnaire (FIQ).

More recently, Tirelli et al. [113] evaluated the effectiveness of O_2_O_3_ autohemotransfusion therapy for the management of fibromyalgia symptoms. The authors included 65 patients with a diagnosis of fibromyalgia according to the definition of the American College of Rheumatology [114]; 55 of them were treated with O_2_O_3_ autohemotransfusion and 10, with O_2_O_3_ rectal insufflations, twice a week for one month and then twice a month as maintenance therapy. There was a significant improvement in terms of pain and fatigue in 45 patients and no adverse events occurred during the whole study, suggesting that O_2_O_3_ therapy could be considered as a safe complementary/integrative treatment in patients affected by fibromyalgia.

Similarly, Moreno-Fernàndez et al. [59] evaluated 20 fibromyalgia patients treated with O_2_O_3_ autohemotransfusion (ten 30–60 μg/mL injections, twice a week) and assessed FIQ, serotonin serum levels, and peripheral mononuclear cell concentrations. Autohemotherapy with O_2_O_3_ showed an important decline in tender points and FIQ score, as well as a decrease in oxidative stress levels, in patients with fibromyalgia.

In conclusion, O_2_O_3_ therapy in different administration modalities (rectal insufflation and autohemotransfusion) could have a role in the complex multidisciplinary management of fibromyalgia patients as a complementary intervention. However, further investigations including wider samples should be carried out in the future to better address this issue.

## 7. Conclusions

In conclusion, this comprehensive review described the state-of-the-art about the mechanisms of action, safety and effectiveness of O_2_O_3_ therapy in the complex management of musculoskeletal disorders in the field of rehabilitation. As previously described, musculoskeletal disorders include several pathological conditions characterized by a consistent prevalence and burden in terms of sanitary costs and residual disability.

In this scenario, O_2_O_3_ therapy might be considered as a promising conservative and mini-invasive intervention with an active role, both alone or in combination with other treatments, in order to reduce pain and improve functioning in patients affected by musculoskeletal disorders. At present, O_2_O_3_ therapy’s efficacy and safety has been mainly investigated in LBP and KOA, as reported by several studies. However, up to now, the evidence about the impact of O_2_O_3_ therapy in other musculoskeletal diseases is still lacking. Thus, further high-quality studies are warranted to deeply understand its therapeutic effects and safety profile in musculoskeletal disorders of PRM interest.

## Figures and Tables

**Figure 1 biomolecules-11-00356-f001:**
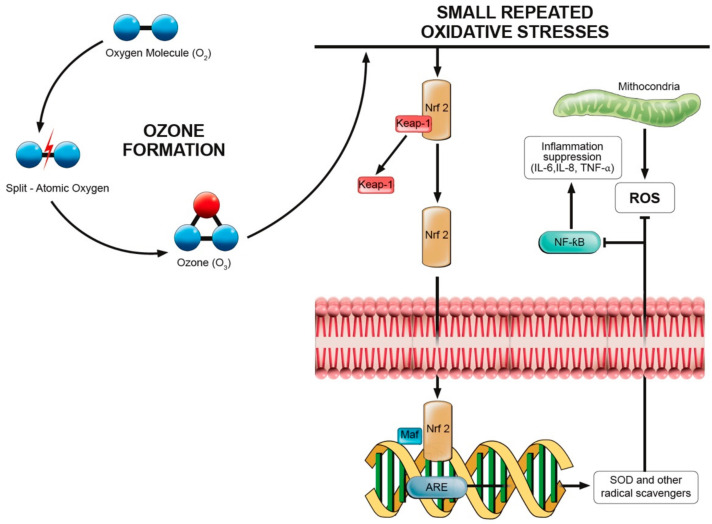
Ozone (O_3_) formation mechanism and intracellular and intranuclear pathways involved in inflammation and oxidative stress.

**Figure 2 biomolecules-11-00356-f002:**
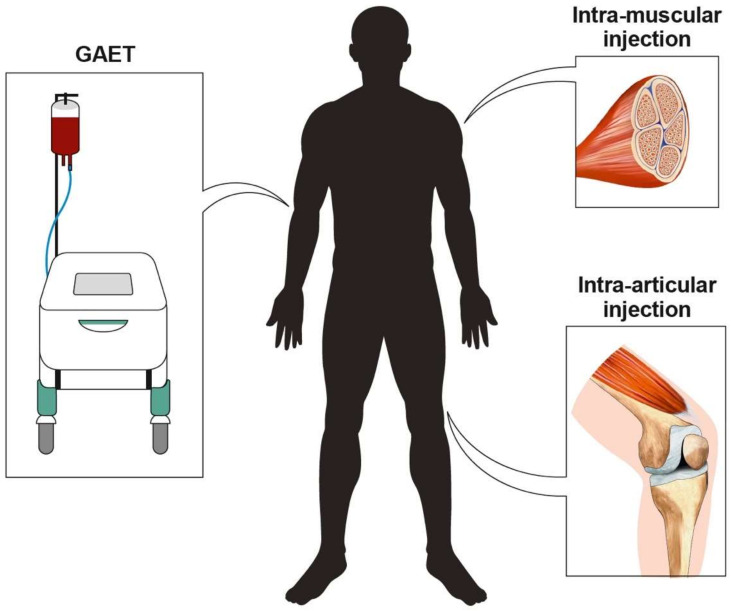
The main means of administration of O_2_O_3_ therapy in the field of rehabilitation.

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
