# Peer review of "Oxygen–Ozone Therapy in the Rehabilitation Field:State of the Art on Mechanisms of Action, Safety andEffectiveness in Patients with Musculoskeletal Disorders"

_biomolecules, 2021, doi:10.3390/biom11030356_

Round 1

Reviewer 1 Report

It is really hard to judge this paper. In my opinion the presented therapy is not well documented in literature. According to Evidence Based Medicine recommendations the level of evidence is relatively weak. 

It means that this area is extremely significant and interesting both for scientists and potential readers. However, we all need profesional and well-written metaanalyses and/or systematic reviews...not the common review papers. The submitted manuscript is good, but for students at University as teaching material, not research work with high quality (PRISMA or PROSPERO registration, data extration, literature search strategy with flow chart etc. etc.)

I think the Biomolecules is a well-known, good and professional journal. There is no space for such review papers (rather for teaching process, it is not science).

If the authors prepare the well conducted metaanalysis or/and systematic review, there will be completely different story...

Author Response

Dear Reviewer 1,

We are aware that the Journal’s standard is high. Thus, we are extremely delighted for the invitation to publish a manuscript on a such important and leading Journal in the field of biogenic substances and their interaction with biological systems.

We would kindly remind the reviewer that the topic, the study design (narrative review), the title and the abstract of this manuscript have been approved and endorsed by the Section Managing Editor of the Journal before the final submission.

We partially agree with the reviewer that the few trials in literature about the efficacy of oxygen-ozone therapy hinder to perform any kind of systematic review or meta-analysis. However, oxygen-ozone is a mini-invasive, safe and low-cost intervention and it is widely used in the common clinical practice in the complex treatment of musculoskeletal disorders in the rehabilitation setting. It is supported by several findings as showed by the present comprehensive review (more than 100 references). Moreover, in the recent years there has been a growing interest in this technique as evidenced by the increase in the number of papers in literature focusing on oxygen-ozone therapy in several pathological conditions, in particular muscoloskeletal disorders. In this context,  it seems constructive to sensitize readers, and in particular those involved in the rehabilitation field, about the potential use of this technique in the complex management of muscoloskeletal disorder, focusing on

the mechanisms of action, safety profile and effectiveness.

We have decided not to perform a systematic review for several reasons that are listed below:

1) the high clinical heterogeneity of the pathological conditions (low back pain, neck pain, knee osteoarthritis, fibromyalgia), the interventions (different treatment plan in terms of sessions and dosages), and the outcomes (pain, functionality, QoL). Thus, we followed the suggestion of the Cochrane Handbook for Systematic Reviews of Intervention (Ver. 6.1, 2020):  "Meta-analysis should only be considered when a group of studies is sufficiently homogeneous in terms of participants, interventions and outcomes";

2) the few high-quality trials in literature on this topic hinder to provide any strong recommendation about oxygen-ozone use. Thus, in the Conclusions we only affirmed “In this scenario O2O3 therapy could be a promising conservative and mini-invasive intervention with an active role in reducing pain and improving functioning in patients affected by musculoskeletal disorders”….and then “However, up to now, evidence about the impact of O2O3 therapy in other musculoskeletal diseases is still lacking; thus, further high-quality studies are warranted to deeply understand its therapeutic effects and safety profile in musculoskeletal disorders of PMR interest.”

3) Oxygen-ozone is a controversial technique which is widely used in common clinical practice to treat several pathological conditions including musculoskeletal disorders. In this context, a comprehensive review focusing on the latest insights about the mechanisms of action, safety profile, treatment protocols and effectiveness could sensitize readers, and in particular those involved in the rehabilitation field, about the evidences and fields of intervention of this technique.

4) The Section Managing Editor of Biomolecules agreed to perform a “comprehensive review” (not a systematic review). Moreover, the title and the abstract have been approved before full manuscript submission.  

We have revised the manuscript following the insightful and constructive comments of the reviewers, providing point-to-point responses below.

Thanking you again for the comment, we hope that you will revise the manuscript from a different perspective.

Reviewer 2 Report

Dear Authors, thank you for your comprehensive review on an interesting subject.

I would recommend you to expand the section "4.Oxygen-ozone therapy and low back pain" adding references to ozone intradiscal and intraforaminal application which has been widely investigated.

Author Response

Dear Authors, thank you for your comprehensive review on an interesting subject.

We would like to thank the reviewer for the positive comment. We are glad that our paper was positively considered.

I would recommend you to expand the section "4. Oxygen-ozone therapy and low back pain" adding references to ozone intradiscal and intraforaminal application which has been widely investigated.

We would like to thank the reviewer for the insightful comment. Accordingly, we modified the “4. Oxygen-ozone therapy for low back pain” Section, dividing it into two different Subsections:

“4.1. Intramuscular paravertebral oxygen-ozone therapy for low back pain” and “4.2. Intradiscal and intraforaminal oxygen-ozone therapy for low back pain”, where we added references on the use of intradiscal and intraforaminal oxygen-ozone therapy in patients suffering low back pain, as suggested by the Reviewer.

Reviewer 3 Report

Comments:

The review is a comprehensive assessment of Oxygen Ozone Therapy Applications in the field of musculoskeletal disorders. It provides the global up-to-date overview of those applications and is an interesting topic in the field of Muskuloskeletal disorders.

The mechanisms of action, safety and effectiveness of O2O3 therapy in the complex scenario of musculoskeletal disorders management- were depicted. Taken together, authors suggested that O2O3 therapy seems to reduce pain and improve function in patients affected by low back pain and knee osteoarthritis.

Major comments

Lines 212-223: explain more how the dosage could be stratified according to the age and other confounding factors that can potentially affect levels of inflammation markers; what is the effective therapeutic window, since too high dose can have opposite effects, are there any therapeutic monitoring assessments available?

In general: to discuss more from the aspects of confounding factors that might affect their results? Has been performed any quality assessment of included papers into the review?

To discuss other alternative therapies that are currently used for the treatment of musculoskeletal disorder instead of O2O3 therapy, to discuss their advantages and disadvantages. Any clinical trials for the combined treatment possibilities?

To discuss more the molecular mechanisms in the particular possibility of generation of free radicals when using such therapy-

Minor comments:

Typo Errors:

Line 81 “proprieties”

Line 202 “2O2O3intramuscular lumbar””…

Some Minor English editing needed

Author Response

The review is a comprehensive assessment of Oxygen Ozone Therapy Applications in the field of musculoskeletal disorders. It provides the global up-to-date overview of those applications and is an interesting topic in the field of Muskuloskeletal disorders.

The mechanisms of action, safety and effectiveness of O2O3 therapy in the complex scenario of musculoskeletal disorders management- were depicted. Taken together, authors suggested that O2O3 therapy seems to reduce pain and improve function in patients affected by low back pain and knee osteoarthritis.

We would like to thank the reviewer for the positive comment.

Major comments

Lines 212-223: explain more how the dosage could be stratified according to the age and other confounding factors that can potentially affect levels of inflammation markers; what is the effective therapeutic window, since too high dose can have opposite effects, are there any therapeutic monitoring assessments available?

We would like to thank the reviewer for the insightful comment. We have expanded in the manuscript the therapeutic window and the dosage issues of oxygen-ozone therapy.  

In general: to discuss more from the aspects of confounding factors that might affect their results? We would like to thank the reviewer for the insightful comment. We have better explained the potential confounding factors, as suggested by the reviewer.

Has been performed any quality assessment of included papers into the review?

We would like to thank the reviewer for the comment. However, we performed a comprehensive review and not a systematic review to focus the reader’s attention about the possible use of oxygen-ozone therapy in different conditions in the rehabilitation setting. Unfortunately, studies in literature addressing this topic are characterized by  a high methodological heterogeneity and focus on different pathological  conditions (low back pain, neck pain, knee osteoarthritis, fibromyalgia), with different therapeutic protocols and outcomes (pain, functionality, QoL). In light of these considerations we did not perform a quality assessment of the studies.

To discuss other alternative therapies that are currently used for the treatment of musculoskeletal disorder instead of O2O3 therapy, to discuss their advantages and disadvantages. Any clinical trials for the combined treatment possibilities?

We would like to thank the reviewer for this suggestion. In the text, we cited a paper performed by Giombini et al. in 2016. The authors performed a combined approach with intra-articular injection of O2O3 and hyaluronic acid in knee osteoarthritis patients. Furthermore, Bellomo et al. in 2020 evaluated the effects of percutaneous injections of O2O3 and water rehabilitation in a woman affected by low back pain.. Moreover, we added another paper in the new subparagraph “4.2. Intradiscal and intraforaminal oxygen-ozone therapy for low back pain”, describing the state-of-art on the use of intradiscal and intraforaminal O2O3 in patients with low back pain. More in detail, we added a paper performed by Gallucci et al. in 2007 that compared a combined treatment with intradiscal and intraforaminal injections of steroid and oxygen-ozone therapy versus steroid only in patients affected by sciatalgia. Lastly, in the conclusion section we have better expanded the concept that O2O3 therapy might be considered a promising conservative and mini-invasive intervention in patients affected by musculoskeletal disorders, both alone or combined with other treatments.

To discuss more the molecular mechanisms in the particular possibility of generation of free radicals when using such therapy-

We would like to thank the reviewer for the insightful comment. We have better discussed the molecular mechanisms underpinning oxygen-ozone therapeutic effect and the potential issue of free radicals generation when using this therapy, as suggested by the reviewer.

Minor comments:

Typo Errors:

Line 81 “proprieties”

We would like to thank the reviewer for the comment. We have corrected the typo.

Line 202 “2O2O3intramuscular lumbar””…

We would like to thank the reviewer for the comment. We have corrected the typo.

Some Minor English editing needed

A complete English revision has been performed.

Round 2

Reviewer 3 Report

The manuscript is nicely revised and includes reviewers` suggestions properly enough.